# Impacts of Climate Change on the Habitat Suitability and Natural Product Accumulation of the Medicinal Plant *Sophora alopecuroides* L. Based on the MaxEnt Model

**DOI:** 10.3390/plants13111424

**Published:** 2024-05-21

**Authors:** Wenwen Rong, Xiang Huang, Shanchao Hu, Xingxin Zhang, Ping Jiang, Panxin Niu, Jinjuan Su, Mei Wang, Guangming Chu

**Affiliations:** 1Agricultural College, Shihezi University, Shihezi 832003, China; 20212012049@stu.shzu.edu.cn (W.R.); shzujp@163.com (P.J.);; 2College of Grassland Science, Xinjiang Agricultural University, Urumqi 830052, China; zhangxingxinshz@163.com

**Keywords:** *Sophora alopecuroides* L., environmental factor, human activities, habitat, natural products

## Abstract

*Sophora alopecuroides* L., a perennial herb in the arid and semi-arid regions of northwest China, has the ecological functions of windbreaking and sand fixation and high medicinal value. In recent years, global warming and human activities have led to changes in suitable habitats for *S. alopecuroides*, which may affect the accumulation of natural products. In this study, MaxEnt 3.4 and ArcGIS 10.4 software were used to predict the distribution of potentially suitable habitats for *S. alopecuroides* in China under climate change. Furthermore, the geographical distribution of *S. alopecuroides* as affected by human activities, the differences in the content of natural products of *S. alopecuroides* between different suitable habitats, and the correlation between natural products and environmental factors were analyzed. The results showed that suitable habitats for *S. alopecuroides* were projected to expand in the future, and the major environmental factors were temperature (Bio1), rainfall (Bio18), and soil pH (pH). When Bio1, Bio18, and pH were 8.4283 °C, 7.1968 mm, and 9.9331, respectively, the distribution probability (*P*) of *S. alopecuroides* was the highest. After adding a human activity factor, the accuracy of the model prediction results was improved, and the area of suitable habitats was greatly reduced, showing a fragmented pattern. Meanwhile, habitat suitability had a specific effect on the content of natural products in *S. alopecuroides*. Specifically, the content of natural products in *S. alopecuroides* in wild habitats was higher than that in artificial cultivation, and highly suitable habitats showed higher contents than those in non-highly suitable habitats. The contents of total alkaloids and total flavonoids were positively correlated with human activities and negatively correlated with land use types. Among them, total alkaloids were negatively correlated with aspect, and total flavonoids were positively correlated with aspect. In addition, it is suggested that Xinjiang should be the priority planting area for *S. alopecuroides* in China, and priority should be given to protection measures in the Alashan area. Overall, this study provides an important foundation for the determination of priority planting areas and resource protection for *S. alopecuroides*.

## 1. Introduction

At present, with global climate change and increased human interference, desertification has accelerated significantly in arid and semi-arid areas, resulting in the migration and degradation of many medicinal plants’ habitats [1,2,3] and even some species populations rapidly declining [4,5,6]. Medicinal plants, as fundamental components of traditional Chinese medicine, play a huge role in the battle against diseases. The growth and distribution of medicinal plants are highly correlated with the growth environment. Habitat has a significant effect on the accumulation of natural products in medicinal plants [7,8,9]. Previous studies have found that plants grown in highly suitable habitats have higher contents of natural products [2,10,11], and the content of natural products directly determines the quality of medicinal plants [12]. Climate warming, environmental degradation, and human interference have led to a decline in the quality of many medicinal plants [8,13].

*Sophora alopecuroides* L. is a perennial plant of the legume family that is widely distributed in West and Central Asia. In China, it is mostly distributed in arid and semi-arid areas such as Xinjiang, Ningxia, Gansu, and Inner Mongolia [14,15]. *S. alopecuroides* is an important medicinal plant widely used in pharmaceutical preparations, such as Kexieling tablets, fuyan suppositories, S. alopecuroides oil liniment, and so on [16]. The whole plant can be used as medicine, with antitumor [17,18,19], anti-inflammatory [20,21], antiviral [22], insecticidal [15,23], and other pharmacological effects. Studies have shown that *S. alopecuroides* has a well-developed root system and strong drought and salinity resistance [15,24] and can fix nitrogen [25]. Therefore, it can improve soil quality [26] and plays an important role in desertification control. In addition, *S. alopecuroides* has been widely used as a botanical pesticide [27,28], green manure [24,29], forage, etc., in agricultural production. Temperature, rainfall, salinity, and other environmental factors have an impact on the distribution and growth of *S. alopecuroides*. Wang [30]’s study found that the annual average temperature, annual variation range for temperature, average temperature of the driest quarter, precipitation of the warmest quarter, and altitude are important factors affecting the distribution of *S. alopecuroides*. Yao [31]’s study found that the main ecological factors affecting the distribution of *S. alopecuroides* are altitude and precipitation in the coldest season. However, due to increased market demand and habitat degradation, wild *S. alopecuroides* populations are rapidly declining in many regions [32]. As a result, *S. alopecuroides* has been cultivated artificially at the small scale in many provinces of China, especially Ningxia, where it has been listed as a daodi medicinal material [33].

Species distribution models are important tools for determining potentially suitable habitats for species [34], of which MaxEnt is currently the most widely used model [35,36]. Many studies have demonstrated the reliability of MaxEnt in analyses of habitat suitability [37,38,39], the effects of climate warming on species distribution [40,41,42], the distribution of invasive and endangered species [43,44,45], and infectious disease risk prediction [46,47,48]. Over the past few years, MaxEnt has also been widely used to predict suitable areas for medicinal plants. For example, Chandora et al. [49] used MaxEnt to predict suitable areas for *Fritillaria cirrhosa* and found that the suitable habitat area for *F. cirrhosa* first increased and then decreased under climate change. Li et al. [10] found that there were differences in the effects of climate warming on suitable areas for the three species of *Coptis* Salisb., and factors such as climate and soil conditions greatly affected their alkaloid content. Yang et al. [50] used the MaxEnt model to analyze the distribution range of *Zanthoxylum nitidum* and found that the distribution range of *Z. nitidum* showed a decreasing trend in future periods and the content of nitidine chloride in suitable habitats was higher than that in unsuitable habitats for *Z. nitidum*. At present, most of the research on *S. alopecuroides* focuses on its chemical composition, pharmacological activity, stress physiology, etc. However, the effects of global climate warming and human activities on the geographical distribution of *S. alopecuroides* are unclear. Furthermore, the correlation between habitat suitability and the natural products of *S. alopecuroides* also needs to be further explored.

In this study, the MaxEnt model and ArcGIS software (ArcGIS software developed by ESRI, a comprehensive geographic information system construction and application platform, referred to as GIS) were used to predict the distribution of potentially suitable areas for *S. alopecuroides* under climate change and human interference in China, and the correlations between habitat suitability, natural product content, and environmental factors were analyzed. The objectives were to identify (1) the dominant environmental factors affecting the geographical distribution and natural product content of *S. alopecuroides* and (2) the relationship between habitat suitability and secondary metabolite content. This study hypothesized that (1) global warming might benefit *S. alopecuroides* growth and expand its distribution; (2) human interference might have a negative impact on the geographical distribution of *S. alopecuroides*; (3) highly suitable areas might be more conducive to the accumulation of natural products of *S. alopecuroides* compared to moderately suitable and less suitable areas. This study will help us to rationally respond to the distribution changes in *S. alopecuroides* caused by climate change and human activities and also help to formulate strategies to improve the medicinal value of *S. alopecuroides*.

## 2. Materials and Methods

### 2.1. Data Acquisition and Processing

The study area was located in China (3.85–53.55° N, 73.55–135.08° E), and *S. alopecuroides* is mostly distributed in the northwest of China [51]. The data on *S. alopecuroides*’s distribution in China were obtained through (1) on-site investigation for two years (2021–2023); (2) consulting the China National Herbarium Resource Platform (http://www.nsii.org.cn/), the China Digital Herbarium (http://www.cvh.org.cn/), and the Global Biodiversity Information Platform (http://www.gbif.org/); and (3) reviewing the national and international literature related to *S. alopecuroides*. A total of 298 distribution points for *S. alopecuroides* in China were obtained. This distribution point data were from 1970 to 2023. To avoid the overfitting phenomenon caused by the aggregation effect, 298 distribution data points were imported into the GIS, the point elements were selected, and buffer zones were set to ensure that each grid (5 km × 5 km) had only one distribution point. Finally, 192 valid distribution points were obtained (Figure 1).

### 2.2. Acquisition of the Environmental Data

Abiotic factors were the main factors affecting the distribution of the species. Climate, soil, and other factors largely determined the distribution range and pattern of the species. The study mainly selected climate data, human activity data, soil data, and topographic data. The sources were as follows: (1) Climate data for the present (1970–2000) and the future four periods (2021–2040, 2041–2060, 2061–2080, and 2081–2100) were downloaded from WorldClim (https://www.worldclim.org/) [52]. Each period has 20 years, and the data are the mean of 20 years. The spatial resolution of all the climate data is 2.5 min. Among them, climate data for the future periods were selected from Beijing Climate Center Climate System Model version 2, Medium-Resolution (BCC-CSM2-MR) and the shared socioeconomic pathway (SSP245) in the Coupled Model Intercomparison Project Phase 6 (CMIP6); (2) Soil data were collected from the National Tibetan Plateau Data Center (http://data.tpdc.ac.cn). Their spatial resolution is 10 m; (3) Topography data were collected using the following method: Elevation data were downloaded from WorldClim and imported into the GIS, the slope tool in the spatial analysis module was used to extract the slope layer “Slope” from the target digital elevation model data, and the aspect tool was used again to extract the aspect layer “Aspect”. The spatial resolution was 2.5 min, and the coordinate system was WGS-1984 (World Geodetic System-1984 Coordinate System); (4) Human footprint data were collected from the International Geoscience Information Network Center (https://sedac.ciesin.columbia.edu), with a spatial resolution of 1 km. These data included eight variables (electric power infrastructure, population density, built-up environments, pasture lands, farmlands, roads, railways, and navigable waterways), which could objectively and comprehensively reflect the intensity and spatial distribution of human activities. Finally, 36 independent environmental variables were selected for the study (Appendix A). The 36 independent environmental variables were imported into the GIS to unify the environmental configuration, a data management tool was used to perform batch resampling commands to unify the resolution of the data from various sources, and the WGS-1984 coordinate system was used. Through the conversion tool, all 36 environment variables were converted into “.asc” format.

To avoid overfitting caused by the collinearity between environmental variables, Spearman’s analysis [53] and the Jackknife method were used to determine the correlation and contribution rate to filter the environmental variables. Variables with a Spearman’s coefficient (Appendix A) less than 0.75 and a contribution greater than 0.5% were retained [54]. For any two variables, Spearman’s coefficient was less than 0.75, so those with a higher contribution rate were retained and those with a lower contribution rate were discarded [55]. Finally, environmental variables with low correlation and a high contribution rate were retained.

### 2.3. Model Settings and Assessment

The coordinates of the 192 distribution points of *S. alopecuroides* and the selected environmental variables were imported into the MaxEnt model, and the coordinate system was WGS-1984. A total of 75% of the data was randomly selected as the training set, and the remainder was used as the validation set. The number of iterations was 1000, the other parameters were the defaults, and the model was run 5 times. In the study, the contribution percentage and the permutation importance of each variable were calculated using the Jackknife method [50]. The accuracy of the model was evaluated by the area under the curve (AUC) of the receiver operating characteristic (ROC) [56]. AUC values range between 0 and 1. The model prediction results are more accurate and the confidence higher when the AUC value is closer to 1 [57]. According to the MaxEnt model, the distribution probability (*P*) of *S. alopecuroides* in China was output. Using the Reclassify command in ArcToolbox in the GIS, the study area was divided into four categories: unsuitable areas (NSAs) (*P* < 0.1), less suitable areas (LSAs) (0.1 ≤ *P* < 0.3), moderately suitable areas (MSAs) (0.3 ≤ *P* < 0.5), and highly suitable areas (HSAs) (0.5 ≤ *P* ≤ 1).

### 2.4. Analysis of the Pharmacologically Active Component Content

Pods from a total of 23 wild populations from Xinjiang, Inner Mongolia, Gansu, Ningxia, and Shaanxi were collected. In each *S. alopecuroides* population, 12 mature pods with healthy growth and no disease were selected, with a total of 276 individuals. In order to avoid the influence of maternal inheritance [58], the sampling distance between individual plants was greater than 10 m, and the sampling distance between populations was greater than 50 km. In addition, 15 mature pods from cultivated *S. alopecuroides* populations in Xinjiang, Gansu, Inner Mongolia, and Ningxia were collected, including 5 from Xinjiang, 3 from Gansu, 3 from Inner Mongolia, and 4 from Ningxia. Finally, a total of 38 samples of *S. alopecuroides* were collected (Table 1).

The seed shells of 12 pods in each population were removed and mixed to obtain clean seed samples. The samples were dried at 60 °C to constant weight, crushed, and passed through an 80-mesh sieve. Then, 0.1 g of the samples was weighed, and 1 mL of extract (prepared with dichloromethane, methanol, and extract (40:10:1)) was added. After homogenization and grinding, they were bathed in water at 80 °C for 1 h. After cooling them to room temperature, centrifugation was conducted at 12,000 rpm for 10 min at room temperature, and the supernatant was used to determine the total alkaloids. At the same time, 0.03 g of the dried samples was weighed, added to 1.5 mL of 60% ethanol, shaken for 2 h at 60 °C, and centrifuged at 12,000 rpm at 25 °C for 10 min, and the supernatant was used to determine the content of total flavonoids. The content was determined using the Total Alkaloid Content Assay Kit and the Total Flavonoid Kit (Grace Biotechnology Co., Ltd., Suzhou, China), respectively.

### 2.5. Redundancy Analysis (RDA) and Cluster Analysis

Redundancy analysis (RDA) can reflect the correlation between response variables and explanatory variables. It is a ranking method that combines regression analysis with principal component analysis and has been widely used in ecological and soil science research [59]. The 17 environmental factors involved in the modeling were used as explanatory variables and the contents of total alkaloids and total flavonoids in the 23 wild *S. alopecuroides* samples were used as response variables for the redundant analysis using Canoco 5.0 software (http://canoco5.com/) to determine the correlation between the environmental factors and the content of medicinal ingredients.

Cluster analysis classifies research objects according to certain indicators and is a common statistical analysis method [60]. The medicinal ingredient content of the 23 wild *S. alopecuroides* samples was clustered using the Origin 2021 software (https://www.originlab.com/) to construct a dendrogram.

The general experimental design and process of this study are shown in Figure 2.

## 3. Results

### 3.1. Model Accuracy Evaluation and Environmental Variable Analysis

The geographical distribution of *S. alopecuroides* in the present and future periods was predicted. Two studies were conducted on the prediction of the current period: with and without human interference. For the current prediction, the AUC value was 0.925 when human interference was not added (Appendix A). After adding human interference, the AUC value was 0.942 (Appendix A). For the future prediction, the AUC value was in the range of 0.925 to 0.927. The AUC value was above 0.9, which was considered to be very accurate.

Among the 36 independent environmental factors, 17 environmental factors were selected according to the conditions that Spearman’s coefficient was less than 0.75 and their contribution rate was greater than 0.5%, which were Bio1, Bio4, Bio6, Bio11, Bio12, Bio15, Bio18, SI, GRAV, pH, SOM, POR, LC, Elevation, Aspect, Slope, and HF (Table 2). In addition to human interference, temperature, precipitation, and soil factors were the dominant environmental factors affecting the distribution probability (*P*) of *S. alopecuroides* among the 17 environmental factors. The permutation importance values and percentage contribution rates (Appendix A) of Bio1, Bio18, and pH were higher than those of the other environmental variables, showing a cumulative permutation importance of 57.1% and a cumulative contribution of 44.7%. The environment variable with the highest gain when used in isolation was Bio1, and *P* of *S. alopecuroides* increased first and then decreased with an increase in Bio1. When Bio1 was in the range of 5.80~11.33 °C, *P* was higher (when Bio1 reached 8.43 °C, the *P* peaked). *P* decreased with an increase in Bio18. When Bio18 was in the range of 7.20~175.39 mm, *P* was higher (*P* was the highest when Bio18 was 7.20 mm). When Bio18 was less than 7.20 mm, *P* no longer changed. It was also found that *P* increased with an increase in the soil pH. When the soil pH was in the range of 8.18~9.93, *P* was higher (*P* was the highest when the soil pH was 9.93). When the soil pH was greater than 9.93, *P* stabilized (Figure 3).

### 3.2. Prediction of the Distribution under the Current Climate Scenario (1970–2000)

The prediction results without human interference showed that *S. alopecuroides* was mainly distributed in northern China, especially Xinjiang, Ningxia, Gansu, Shaanxi, Shanxi, and Inner Mongolia (Figure 4A). The suitable area for *S. alopecuroides* was distributed radially as a whole. The HSAs were at the center, surrounded by the MSAs, and the LSAs were distributed at the edge of the MSAs, which were degressive layer by layer. Among them, the HSAs were mainly distributed in Xinjiang (western Altay, western Tacheng, Ili River Valley, Bortala, the southern and northern foothills of the Tianshan Mountains, and oases at the western and northern edges of the Tarim Basin), Ningxia (Shizuishan, Yinchuan, Wuzhong, and northern Zhongwei), Gansu (north–central Baiyin), and Inner Mongolia (southern Bayanzhuoer). The MSAs were mainly distributed in Gansu (Lanzhou, southern Baiyin, central Tianshui, Pingliang, and Qingyang), Shaanxi (Yulin, Yan’an, and Baoji), Shanxi (Linfen, Yuncheng, and western Jinzhong), Inner Mongolia (Ordos), and Xinjiang. The LSAs were mainly distributed in Xinjiang (Altay, Hami, and Turpan), Gansu (Jiuquan and Dingxi), Inner Mongolia (Alashan, northwest Bayanzhuoer, Baotou, Hohhot, Ulanqab, southwest Xilin Gol, and Chifeng), Shaanxi (Weinan, Hanzhong, and Ankang), and southeast and northwest Shanxi.

The prediction results for human interference (Figure 4B) showed that the MSAs and HSAs of *S. alopecuroides* were mainly distributed in Xinjiang, Ningxia, and the surrounding areas of Ningxia, and the total suitable area for *S. alopecuroides* was greatly reduced. Without human interference, the total suitable areas for *S. alopecuroides* were about 2,805,829.56 km^2^, accounting for about 29.23% of the total area of China (the study area); the HSAs were about 423,826.62 km^2^, accounting for about 4.42%; the MSAs were about 751,680.34 km^2^, accounting for 7.83%; and the LSAs were about 1,630,322.60 km^2^, accounting for 16.99%. With human interference, the total suitable areas for *S. alopecuroides* were about 2,036,474.912 km^2^, accounting for 21.22% of the total area of the study area; the HSAs, MSAs, and LSAs covered 222,649.47 km^2^, 436,661.68 km^2^, and 1,377,163.76 km^2^, accounting for 2.32%, 4.55%, and 14.35% of the study area, respectively (Figure 4C).

With human interference, the suitable area for *S. alopecuroides* decreased seriously. Western Inner Mongolia transitioned from LSAs to NSAs; central and southern Inner Mongolia transitioned from HSAs and MSAs to LSAs; northern Shaanxi transitioned from MSAs to LSAs; and the margins of the HSAs in Xinjiang and Ningxia transitioned to MSAs. In general, the proportions of the HSAs, MSAs, and LSAs showed a decreasing trend. The HSAs decreased the most seriously, and the HSAs in each province decreased in the range of 41.46~75.08%. The area reduction percentage of the HSAs in Inner Mongolia was the largest, with an area reduction of 42,959 km^2^. The decrease in the HSAs in Gansu was the smallest, with an area reduction of 16,044 km^2^. Although the percentage of area reduction of the HSAs in Xinjiang was not the largest, it was the largest reduction in area, which was 115,216 km^2^ (Figure 4D).

### 3.3. Prediction Distribution under Future Climate Scenarios

The results of the prediction (Figure 5A) showed the MSAs and HSAs for *S. alopecuroides* were mainly distributed in Xinjiang (western Altay, western Tacheng, Ili River Valley, Bortala, the southern and northern foothills of the Tianshan Mountains, central Hami, and oases at the western and northern margins of the Tarim Basin), Gansu (Baiyin, Lanzhou, Qingyang), Ningxia, Shaanxi (Yulin and Yan’an), Shanxi (Yuncheng and Linfen), and Inner Mongolia (Ordos). In the future four periods (Figure 5B), the suitable area for *S. alopecuroides* expanded slightly and showed a trend of decreasing first and then increasing. During the period of 2041–2060, the total suitable area was the smallest, and the MSAs and LSAs decreased (Appendix A). The HSAs, MSAs, and LSAs covered 490,742.85 km^2^, 735,935.35 km^2^, and 1,903,712.96 km^2^, accounting for 5.11%, 7.67%, and 19.83% of the study area, respectively. During the period of 2061–2080, the total suitable area was the largest, and the HSAs, MSAs, and LSAs increased (Appendix A). The HSAs, MSAs, and LSAs covered 554,009.10 km^2^, 904,549.93 km^2^, and 2,035,326.80 km^2^, accounting for 5.77%, 9.42%, and 21.21% of the study area, respectively. In the four periods, the HSAs were largest in Xinjiang Province, followed by Inner Mongolia, Gansu, Ningxia, and Shaanxi. The HSAs in each province basically increased year by year (Figure 5C).

### 3.4. Analysis of Alkaloid and Flavonoid Content in S. alopecuroides in Different Habitats

A total of 38 samples of *S. alopecuroides* were collected, including 23 wild samples and 15 cultivated samples (Figure 6A). Among the 23 wild *S. alopecuroides* samples (K1 to K12), the content of total alkaloids ranged from 59.60 mg/g to 98.99 mg/g, and the content of total flavonoids ranged from 6.01 mg/g to 11.29 mg/g. The total alkaloid content of K5 (98.99 mg/g) was the highest, while that of K2 (59.60 mg/g) was the lowest. The content of total flavonoids of K7 (11.29 mg/g) was highest, while that of K17 (6.01 mg/g) was the lowest. In addition, among the 15 cultivated *S. alopecuroides* samples (K24-1 to K27-4), the total alkaloid content was between 59.64 mg/g and 80.73 mg/g, and the total flavonoid content was in the range of 6.68 mg/g to 10.69 mg/g. The total alkaloid content of K24-2 (80.73 mg/g) was highest, while that of K24-3 (59.64 mg/g) was the lowest. The total flavonoid content of K24-5 (10.69 mg/g) was highest, while that of K26-3 (6.68 mg/g) was the lowest.

Comparing the wild samples and the cultivated samples from the same area, it was found that the contents of K24-1 (74.12 mg/g) were lower than those of K11 (84.01 mg/g), those of K24-4 (65.87 mg/g) were lower than those of K4 (68.89 mg/g), those of K25-3 (65.44 mg/g) were lower than those of K19 (69.70 mg/g), and those of K26-1 and K26-2 (64.19 mg/g and 64.83 mg/g) were lower than those of K15 (69.49 mg/g). More importantly, the average contents of total alkaloids (76.65 mg/g) and total flavonoids (8.42 mg/g) in the 12 wild samples K1~K12 in Xinjiang were the highest in the wild samples, and the average contents of total alkaloids (69.56 mg/g) and total flavonoids (8.93 mg/g) in the 5 cultivated samples K24-1~K24-5 in Xinjiang were also the highest in the cultivated samples.

Among the 38 samples of *S. alopecuroides*, the average contents of total alkaloids and total flavonoids in the 23 wild samples were higher than those in the 15 cultivated samples (Figure 6B). The average content of total alkaloids in the 15 cultivated samples was 65.68 mg/g. Of the 23 wild *S. alopecuroides* samples, the average content of total alkaloids in the 10 samples of *S. alopecuroides* distributed in the HSAs was 77.90 mg/g, and the average content of total alkaloids in the 13 wild samples distributed in the non-HSAs was 69.25 mg/g. In addition, the average content of total flavonoids in the 15 cultivated samples was 7.92 mg/g. The average contents of total flavonoids in the wild samples from the 10 HSAs and 13 non-HSAs were 8.07 mg/g and 7.91 mg/g, respectively. It is worth noting that in the wild samples, the average content of total alkaloids and total flavonoids in the HSAs was higher than that in the non-HSAs.

Among the 17 environmental factors involved in modeling, Bio1, Bio4, GRAV, SOM, SI, pH, Slope, Aspect, and HF were positively correlated with the total alkaloid content of *S. alopecuroides* (Aspect > GRAV > Bio4 > HF > pH > SI > Slope > SOM > Bio1). Bio6, Bio11, Bio12, Bio15, Bio18, POR, LC, and Elevation were negatively correlated with the total alkaloid content of *S. alopecuroides* (LC > POR > Bio15 > Elevation > Bio11 > Bio18 > Bio6 > Bio12). Bio1, Bio6, Bio11, GRAV, SOM, POR, SI, Slope, and HF were positively correlated with the total flavonoid content (HF > SOM > Bio6 > Bio11 > GRAV > SI > Slope > Bio1 > POR). Bio4, Bio12, Bio15, Bio18, LC, pH, Aspect, and Elevation were negatively correlated with the total flavonoid content (LC > Bio12 > Bio18 > Aspect > Bio4 > Elevation > pH > Bio15) (Figure 6C). The cluster analysis of the total alkaloid and total flavonoid contents of 23 wild populations (Figure 6D) found that the 23 samples were clustered into three categories when the critical distance was 10. K5 and K6 of the HSAs were clustered into the first category. K1, K8, K9, K21, and K23 of the HSAs and K7, K11, and K18 of the MSAs were clustered into the second category. The remaining was clustered into the third category, most of which were from the MSAs and LSAs. Interestingly, K2, K12, and K20 of the HSAs were also clustered into the third category, and K2 in the second category was found to be separated independently.

## 4. Discussion

In this study, the MaxEnt model was used to simulate the potential suitable areas for *S. alopecuroides* in China, and the main environmental factors were analyzed [61]. The results showed that *S. alopecuroides* was mainly distributed in northwest China, especially in arid areas such as Xinjiang, Ningxia, Gansu, and Inner Mongolia, which is consistent with the results of a previous germplasm resource survey [62,63]. Previous studies have shown that environmental factors such as temperature, precipitation, and soil pH have a certain impact on the growth and distribution of medicinal plants [64,65]. In this study, annual mean temperature (Bio1) had the highest contribution rate (16.8%) among the 17 environmental variables involved in the modeling, followed by the precipitation of the warmest quarter (Bio18, 16.4%) and the soil pH (11.5%). In addition, the environment variable with the highest gain when used in isolation was Bio1, which provided more valid information and had a greater influence on S. alopecuroides’ distribution. This indicates that temperature, precipitation, and soil pH are the dominant environmental factors which affect the distribution of areas suitable for *S. alopecuroides*. The annual mean temperature plays a decisive role in the distribution of the suitable areas for *S. alopecuroides*. An annual average temperature between 5.80 °C and 11.33 °C was most suitable for the growth of *S. alopecuroides*, and when it was 8.43 °C, the distribution probability was the highest. It can be seen that mid-temperate and warm temperate regions are more suitable for the growth of *S. alopecuroides*. Then, when the precipitation of the warmest quarter was less than 175.39 mm and the soil pH was greater than 8.18, this was most suitable for the growth of *S. alopecuroides*, congruent with its ecological features of drought tolerance and salinity tolerance [15,51]. In a word, these findings provide valuable guidance for the selection of planting areas for *S. alopecuroides*.

Global warming greatly affects the distribution of species, and different species respond differently to climate change [10,11]. Previous researchers have not studied the effect of climate warming on the geographical distribution of *S. alopecuroides*. The results of this study showed that under natural conditions, the suitable area for *S. alopecuroides* increased slightly in the future periods, which is similar to the results of studies on *Houttuynia cordata* [66] and *Litsea cubeba* [67]. This shows that *S. alopecuroides* has strong adaptability to climate change. The research showed that the highly suitable area was set to increase in the future, and it was mostly distributed in arid areas such as Xinjiang, Gansu, Ningxia, and Inner Mongolia. The indicates that *S. alopecuroides* is highly suitable for growth in these arid areas. As the most vulnerable regions in the world, arid regions are the focus of global ecological governance [68]. *S. alopecuroides* has a strong ability for drought tolerance and salinity tolerance, and underground buds on rhizome nodes have a strong asexual reproduction ability. Considering its strong survival and adaptability, it can be used as an important plant for desertification control in arid areas.

For the past few years, the aggravation of human activities and the deterioration of the natural environment have led to the loss or the high fragmentation of suitable habitats for medicinal plants. This hinders their normal growth and reproduction, leading to a decline in or even the extinction of medicinal plants [69,70]. In this study, after adding a human interference factor, the AUC values increased, and the prediction accuracy of the model was improved. This indicates that human activities have a strong intervention effect on the geographical distribution of *S. alopecuroides*. Similar results have been found in studies of *Solanum rostratum* [71] and Ranunculaceae species [69]. Furthermore, the interaction of climate and human activity factors can exacerbate ecosystem vulnerability [72]. After adding the human interference factor, the HSAs, MSAs, and LSAs of *S. alopecuroides* decreased and distributed in fragments, among which the HSAs were most affected. This indicates that human activities have a certain negative impact on *S. alopecuroides* growth, and *S. alopecuroides* is very sensitive to the disturbance of human activities. Xu et al. [73] and Cao et al. [74] also found a negative correlation between the strength of human activity and the range of species distribution. In addition, this study found that the area reduction percentage of the HSAs in Inner Mongolia was the largest, the Alxa League transitioned from LSAs into NSAs, and a large proportion of the MSAs in northern Shaanxi became LSAs. The western part of Inner Mongolia is generally a pastoral area, which may tend towards overgrazing, leading to a decline in *S. alopecuroides*’s habitat. Land reclamation in northern Shaanxi is aggravated, and the vegetation coverage of the Loess Plateau itself is diminishing and yet still grazed upon. Considered comprehensively, it should be protected in situ, strict prohibition be implemented, priority protection areas set up, and man-made damage reduced.

The artificial cultivation of medicinal plants has become an important way to meet the growing market demand. Medicinal plant cultivation should pay attention not only to yield but also to the content of its natural products [75,76]. Alkaloids and flavonoids, the main medicinal components of *S. alopecuroides*, have been widely studied and used [77]. Previous studies have shown that medicinal plants growing in HSAs have a higher content of natural products than those from non-HSAs [2,50,78]. In this study, the content of natural products in the *S. alopecuroides* samples from HSAs (especially Tacheng, Aksu, Hami, and other areas of Xinjiang) was higher than in those from other areas, indicating HSAs are conducive to the production and accumulation of secondary metabolites in *S. alopecuroides*. This is consistent with study results on medicinal plants such as *Pistacia chinensis* [10] and *Cordyceps sinensis* [11]. This study found that the average contents of total alkaloids and total flavonoids in wild *S. alopecuroides* were higher than those of cultivated *S. alopecuroides*. Equally, studies on traditional Chinese medicinal plants such as licorice [79] and *Corydalis saxicola* [80] found that the content of natural products of wild plants was higher than that of cultivated plants. This may be due to unscientific cultivation practices (e.g., intercropping, relay intercropping, and water management) and the excessive pursuit of yield (ignoring medicinal value), and studies have found that some environmental stresses (such as drought and salinity stress) in wild habitats may promote the accumulation of secondary metabolites in *S. alopecuroides* [24,32]. Moreover, this study found that the accumulation of medicinal contents in *S. alopecuroides* had a significant correlation with aspect, land type, and human activities. Aspect has an important impact on biodiversity, plant growth and development, productivity, and material accumulation by changing certain environmental factors, such as light, soil moisture, soil temperature, and soil mineral content [81]. These factors should be considered in the selection of planting areas, and the negative impact of human factors should be minimized to control land reclamation. In short, it is likely that HSAs will be the priority planting areas for *S. alopecuroides* in the future, especially in Xinjiang, which is highly suitable for the growth of *S. alopecuroides*, and Xinjiang is the largest province in China. There is powerful potential to use it as a key development area for *S. alopecuroides*. The transformation of *S. alopecuroides* from wild harvesting to planting ensures that the market demand is met without exhausting natural resources and at the same time increases its content of natural products and ensures the stability of production.

## 5. Conclusions

Assessing the effect of climate change and human activities on the geographical distribution of *S. alopecuroides* has a guiding role in the protection and utilization of the medicinal plant *S. alopecuroides*. This study found that the geographical distribution of *S. alopecuroides* was mainly affected by environmental factors such as temperature, rainfall, and soil pH, and the suitable areas for *S. alopecuroides* showed a slight expansion trend in the future. However, the suitable areas were greatly reduced and fragmented when considering the interference of human activities. Equally, the accumulation of natural products in *S. alopecuroides* in highly suitable habitats was higher than that in non-highly suitable habitats, and the content of natural products in wild *S. alopecuroides* was higher than that in cultivated plants. This study provides a reference for habitat evaluation, priority conservation areas, and priority planting areas for *S. alopecuroides* in China. It is recommended to establish nature reserves in the western part of Inner Mongolia and carry out artificial cultivation in Xinjiang. In addition, the metabolic mechanism of *S. alopecuroides* should be further explored at the molecular level to provide reference for increasing its medicinal value.

## Figures and Tables

**Figure 1 plants-13-01424-f001:**
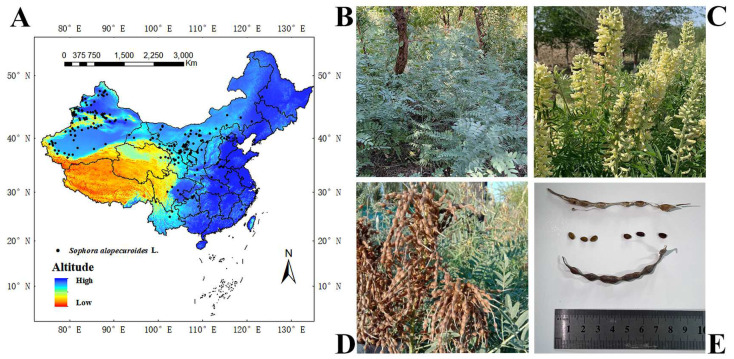
Spatial distribution and growth status of *S. alopecuroides* in China. (**A**) One hundred and ninety-two distribution points, (**B**) vegetative period, (**C**) flowering period, (**D**) fruiting period, and (**E**) seeds.

**Figure 2 plants-13-01424-f002:**
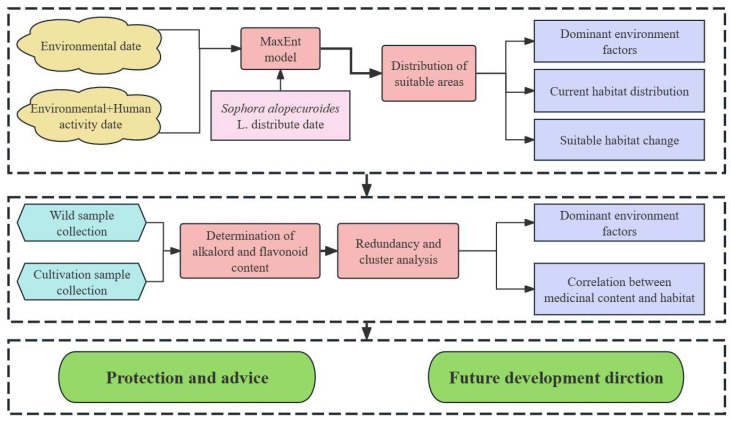
Experimental design and flowchart for this study.

**Figure 3 plants-13-01424-f003:**
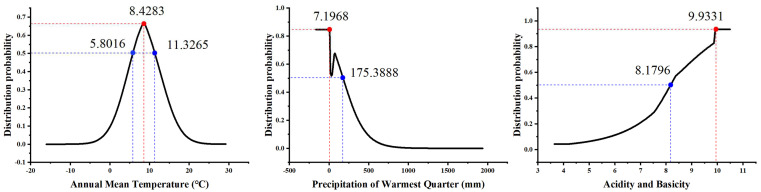
Response curve of distribution probability to main environmental variables.

**Figure 4 plants-13-01424-f004:**
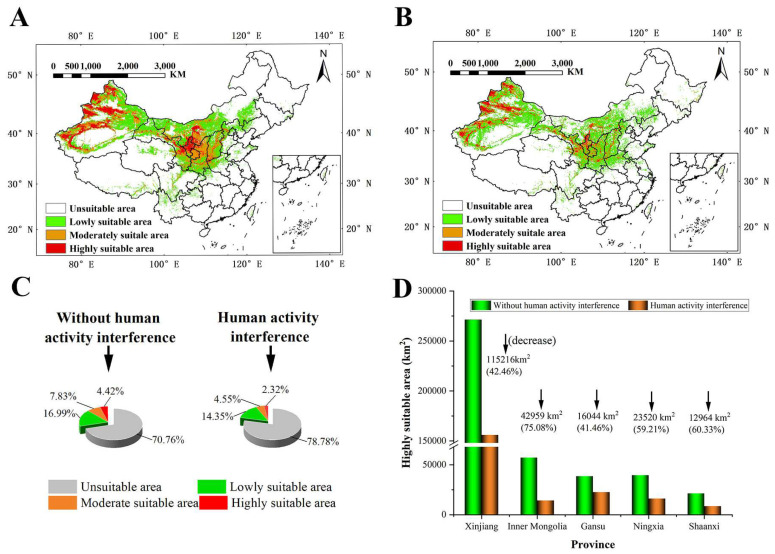
The suitable area for *S. alopecuroides* under the current climate scenario (1970–2000) (**A**) without human interference and (**B**) with human interference; percentage of each suitable area (**C**); and changes in HSAs in each province (**D**).

**Figure 5 plants-13-01424-f005:**
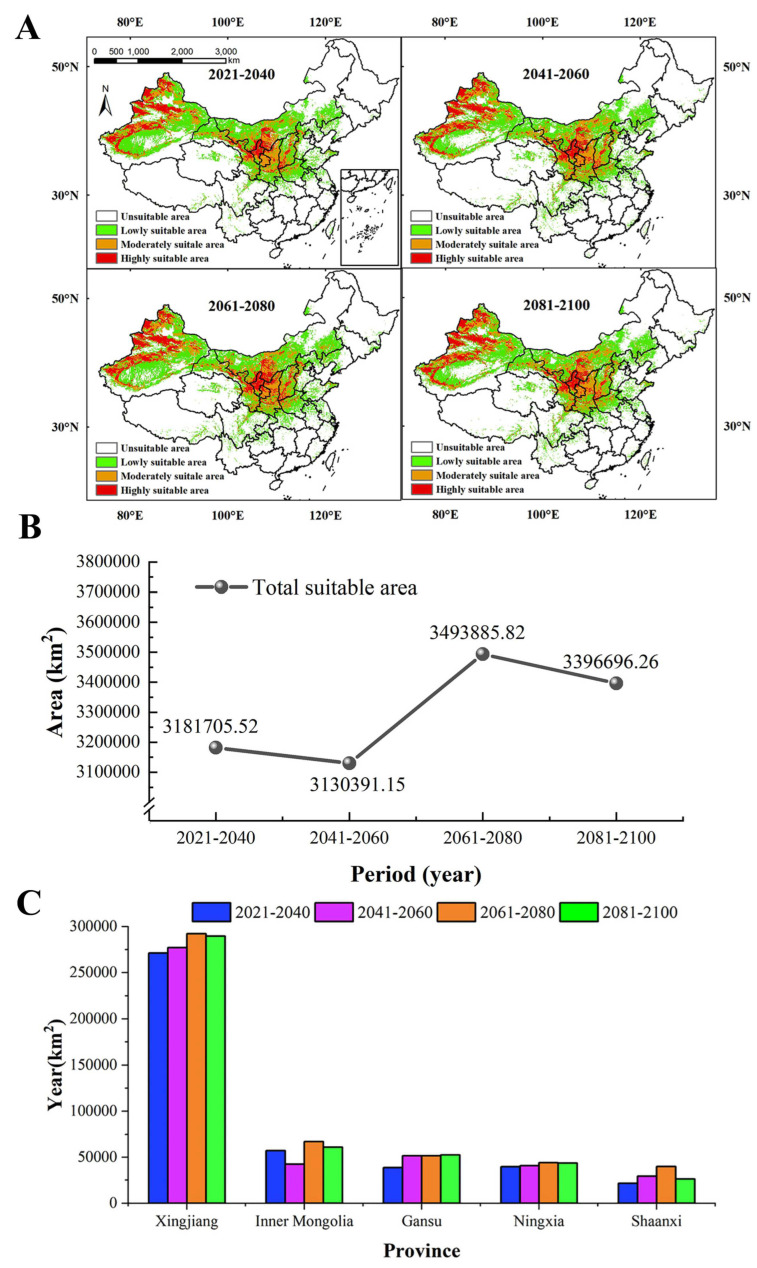
The geographical distribution of *S. alopecuroides* in the future (**A**), the change in the total suitable area (**B**), and the highly suitable areas in each province in the future (**C**).

**Figure 6 plants-13-01424-f006:**
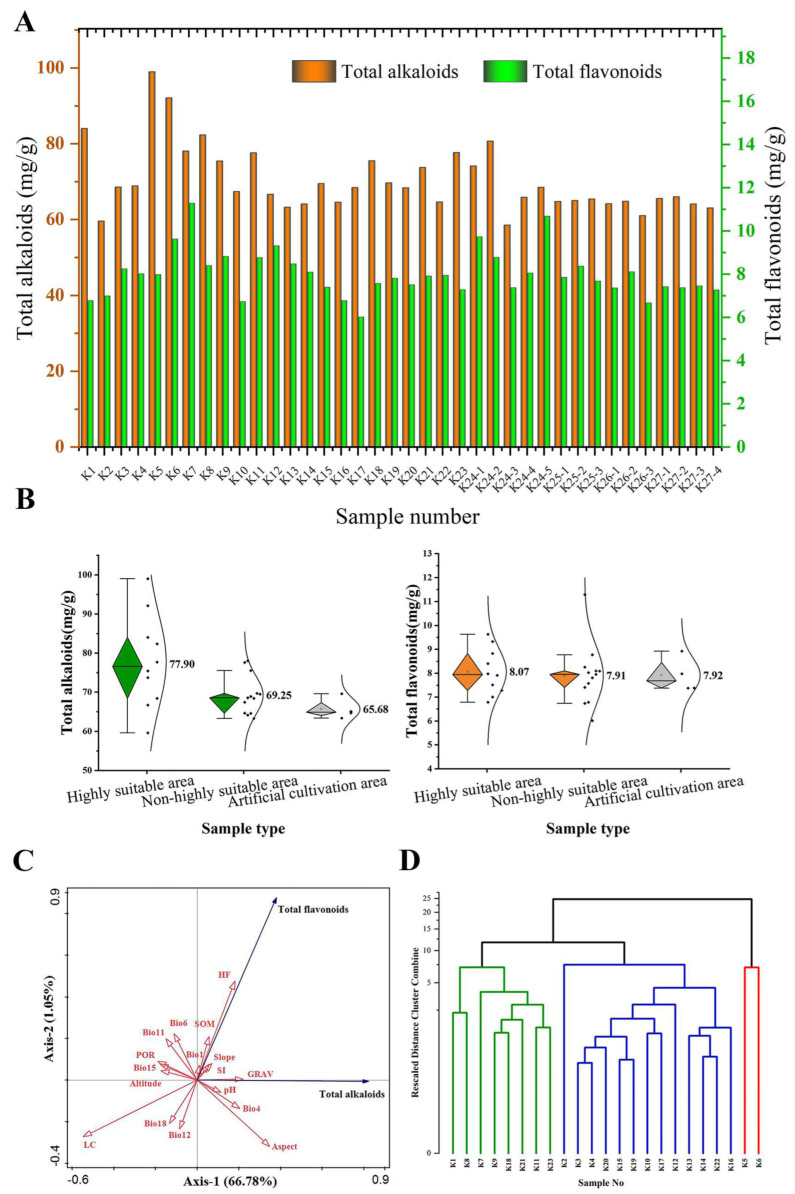
Analysis of total alkaloid and total flavonoid content of *S. alopecuroides*. (**A**) Comparison of total alkaloid and total flavonoid content of *S. alopecuroides* samples in different habitats; (**B**) contents of total alkaloids and total flavonoids of wild and cultivated *S. alopecuroides* samples; (**C**) correlation analysis of total alkaloid and total flavonoids content with environmental factors; (**D**) dendrogram constructed using the total alkaloid and total flavonoid content of 23 *S. alopecuroides* samples.

**Table 1 plants-13-01424-t001:** Location of *S. alopecuroides* pod sampling points.

Sample No	Location	Longitude	Latitude	Altitude (m)	Nature
K1	Shihezi, Xinjiang, China	86.27° E	44.77° N	366	Wild
K2	Yili, Xinjiang, China	81.69° E	43.74° N	712	Wild
K3	Bole, Xinjiang, China	82.34° E	44.81° N	305	Wild
K4	Qitai, Xinjiang, China	90.01° E	43.61° N	1914	Wild
K5	Tacheng, Xinjiang, China	82.01° E	46.19° N	667	Wild
K6	Aksu, Xinjiang, China	80.26° E	41.17° N	1116	Wild
K7	Hami, Xinjiang, China	92.86° E	43.06° N	816	Wild
K8	Korle, Xinjiang, China	86.57° E	42.05° N	1052	Wild
K9	Aletai, Xinjiang, China	87.82° E	47.35° N	513	Wild
K10	Turpan, Xinjiang, China	89.07° E	42.82° N	14	Wild
K11	Hotan, Xinjiang, China	81.69° E	36.81° N	1451	Wild
K12	Kashger, Xinjiang, China	76.05° E	39.40° N	1249	Wild
K13	Bayanzhuoer, Inner Mongolia, China	107.36° E	40.75° N	1060	Wild
K14	Ejin Banner, Inner Mongolia, China	101.04° E	41.96° N	931	Wild
K15	Ordos, Inner Mongolia, China	108.92° E	40.49° N	1102	Wild
K16	Alashan League, Inner Mongolia, China	105.72°E	38.86° N	1608	Wild
K17	Jiuquan, Gansu, China	94.12° E	39.89° N	1312	Wild
K18	Zhangye, Gansu, China	101.11° E	38.79° N	1826	Wild
K19	Wuwei, Gansu, China	103.15° E	38.61° N	1346	Wild
K20	Shizuishan, Ningxia, China	106.54° E	38.72° N	1113	Wild
K21	Wuzhong, Ningxia, China	106.08° E	37.37° N	1359	Wild
K22	Guyuan, Ningxia, China	106.25° E	36.02° N	1738	Wild
K23	Yulin, Shaanxi, China	107.80° E	37.51° N	1408	Wild
K24-1	Hotan, Xinjiang, China	81.47° E	36.61° N	1552	Cultivated
K24-2	Hotan, Xinjiang, China	81.60° E	37.01° N	1365	Cultivated
K24-3	Yili, Xinjiang, China	81.49° E	43.89° N	639	Cultivated
K24-4	Qitai, Xinjiang, China	90.31° E	43.51° N	1907	Cultivated
K24-5	Jimsar, Xinjiang, China	88.65° E	44.21° N	1983	Cultivated
K25-1	Minqin, Gansu, China	103.74° E	38.9° N	1275	Cultivated
K25-2	Lanzhou, Gansu, China	103.25° E	36.34° N	1742	Cultivated
K25-3	Wuwei, Gansu, China	114.63° E	38.83° N	1225	Cultivated
K26-1	Ordos, Inner Mongolia, China	107.92° E	39.49° N	1282	Cultivated
K26-2	Ordos, Inner Mongolia, China	108.92° E	40.49° N	1306	Cultivated
K26-3	Alashan League, Inner Mongolia, China	106.72° E	38.56° N	1628	Cultivated
K27-1	Yanchi, Ningxia, China	107.02° E	38.11° N	1390	Cultivated
K27-2	Yanchi, Ningxia, China	106.98° E	38.37° N	1344	Cultivated
K27-3	Taole, Ningxia, China	106.56° E	38.82° N	1133	Cultivated
K27-4	Zhongwei, Ningxia, China	104.53° E	37.46° N	1659	Cultivated

**Table 2 plants-13-01424-t002:** Seventeen independent environmental variables.

Data Type	Variable	Description	Unit
Climate variables	Bio1	Annual Mean Temperature	°C
	Bio4	Temperature Seasonality	°C
	Bio6	Min Temperature of Coldest Month	°C
	Bio11	Mean Temperature of Coldest Quarter	°C
	Bio12	Annual Precipitation	mm
	Bio15	Precipitation Seasonality	mm
	Bio18	Precipitation of Warmest Quarter	mm
Soil variables	SI	Powder Grains	wt.%
	GRAV	Gravel	wt.%
	pH	Acidity and Basicity	—
	SOM	Organic Matter Content	%
	POR	Porosity	%
	LC	Land Use Type	—
Terrain variables	Elevation	Elevation	m
	Aspect	Aspect	°
	Slope	Slope	°
Human activities	HF	Human Footprint	—

## Data Availability

The data presented in this study are available as Appendix A.

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
