# Peer review of "Impacts of Climate Change on the Habitat Suitability and Natural Product Accumulation of the Medicinal Plant Sophora alopecuroides L. Based on the MaxEnt Model"

_plants, 2024, doi:10.3390/plants13111424_

Round 1

Reviewer 1 Report

Comments and Suggestions for Authors

The manuscript presents a study where MaxEnt and GIS were used to predict the distribution of potentially suitable habitats of S. alopecuroides in China under climate change.

The abstract is clear and well written.

I suggest replacing ArcGIS by GIS and define the acronym because MaxEnt is a method and that's ok, and ArcGIS is a software, not a method. They are different. If you refer to GIS it is clearer in terms of methodology applied. Then in the methodology section refer to what software you used. Also in this section, introduction, add some examples of using GIS in this context.

Ok, now I see that you have the materials and methods section, but the structure is not correct. Please move the materials and methods section to before the results.

Also the study area and period of analysis should be also described and explained in methodology section.

In results section, each environmental factor has associated an acronym. That should be explained in methodology section and also, for instance, you referred “altitude” that I suggested to rename as elevation, and I see that you derived aspect and slope from DEM (supposing that you defined with this term). A lot of information is missing, such as, what is the source of elevation layer (DEM), and of the other samples/data? What was the coordinate system used? How did you generate aspect and slope (I know the answer, but it's not mandatory for the reader to know. So you have to explain all of that. Also a figure with the study zone is important, as well as, a flowchart of the methodology employed.

Miss units in table S2 in supplmentary materils (latitude and longitude).

A lot of acronyms were not defined.

I cannot evaluate section 2.4 Analysis of alkaloids and flavonoids content in S. alopecuroides in different habitats with precision.

The language is very good, the text is simple. The discussion section is very clear with some comparison to other studies. The references are updated also.

The study is very interesting and in my opinion it is worthy of publication; I just have some comments in the pdf file; the main modification should be the structure. Some other details are missing to improve the paper.

More comments in pdf attached.

I can only accept the publication of this work after a major revision.

Author Response

     Thank the reviewer for taking the time to review my manuscript. Your comments have greatly improved our manuscript. The following PDF is my point-to-point reply to the question. Thank you again !

Reviewer 2 Report

Comments and Suggestions for Authors

Reviewer’s Comments:

Title: Impacts of climate change on habitat suitability and natural products accumulation of medicinal plant Sophora alopecuroides L. based on MaxEnt model

The authors have employed MaxEnt and ArcGIS software to study and predict the distribution of potentially suitable habitats of S. alopecuroides in China under climate change. They authors have further estimated total alkaloids and flavonoids content and correlated them with the habitat suitability. The study is well designed and executed. The manuscript (MS) may be recommended for publication after addressing following comments:

1.      The language (mainly usage of words) needs to be checked throughout the MS. For instance:

1.1.  Lines 22-23: distribution probability was ‘highest’, not ‘largest’.

1.2.  Lines 31-32: Sentence needs to be rewritten.

1.3.  Line 110: ‘adding and not adding’ should be replaced by ‘with and without’.

There are similar several words that need to be replaced. Hence, the authors should get the MS checked by a native English speaker or take professional assistance.

2.     Some references are missing. Line 54-55, authors should mention with reference what pharmaceuticals contain Sophora alopecuroides L. Further, reference is missing in Line 58 for N2 fixation.

3.     Abbreviations: Numerous abbreviations have been used (e.g. AUC, HAS, MSA, LC, LSA,…etc.) but none of them have been explained at their first mention in the MS but later in materials and methods. Authors are suggested to rectify this.

4.     Under result section, the parameters ‘Bio1, Bio4, Bio6, Bio11, Bio12, Bio15, Bio18, SI, GRAV, pH, 399 SOM, POR, LC, Altitude, Aspect, Slope and HF’ have been correlated but not defined. It is better if the authors would give a short definition and full form (e.g. GRAV, SI, SOM, etc.) and significance of each of these parameters justifying why only these have been selected for the study.

5.     S. alopecuroides has been reported for alkaloids, flavonoids, flavonoid glycosides, steroids, and polysaccharides. Why did the authors choose to measure only alkaloids and flavonoids?

6.     Along with latitude, longitude and altitude of the 27 sites of China, it is recommended that authors should provide the GPS co-ordinates of the places to provide better clarity to the readers who wish to reproduce or expand the study.

Comments on the Quality of English Language

The quality of English language in manuscript is low and needs to be revised through professional assistance.

Author Response

  Sincerely thank the reviewer for the time and energy paid for my manuscript, your help is of great significance to me. The following PDF is my point-to-point response to the question. Thank you again !

Round 2

Reviewer 1 Report

Comments and Suggestions for Authors

The manuscript presents a study where MaxEnt and GIS were used to predict the distribution of potentially suitable habitats of S. alopecuroides in China under climate change.

The authors improved the manuscript, respecting some of my comments. However, others were not implemented, and I insist that they are relevant to improve even more the manuscript. So, I ask the authors to consider:

This suggestion should be performed in Introduction section (line89): “I suggest replacing ArcGIS by GIS and define the acronym because MaxEnt is a method and that's ok, and ArcGIS is a software, not a method. They are different. If you refer to GIS it is clearer in terms of methodology applied. Then in the methodology section refer to what software you used. Also in this section, introduction, add some examples of using GIS in this context.”

Line 73: I suggest to add this recent study to your references:

Nuno Garcia, João Alírio, Daniel Silva, João C. Campos, Lia Duarte, Salvador Arenas-Castro, Isabel Pôças, Neftali Sillero, and Ana C. Teodoro "MontObEO, Montesinho biodiversity observatory: an Earth observation tool for biodiversity conservation", Proc. SPIE 12734, Earth Resources and Environmental Remote Sensing/GIS Applications XIV, 1273411 (19 October 2023); https://doi.org/10.1117/12.2678524

After that, there are some minor comments that I inserted in the manuscript attached.

Lines 208-214: These part of txt seems to be of MDPI template.

Is there some kind of validation process?

More comments in pdf attached.

I can only accept the publication of this work after a minor revision.

Author Response

  Thank you for your decision and constructive comments on my manuscript. We agree with the reviewers' suggestions and will incorporate the recommended changes into the manuscript. We sincerely appreciate the time and effort invested by the reviewers in evaluating our manuscript.
